# Proposal: A Reinforcement Learning Framework for Protein Function Prediction

**Chenxi Hu**
*2024310688*

**Fei Long**
2024316091

**Renrui Tian**
2024310636

## Abstract

Proteins are the foundation of all life activities. Accurate predicting protein function is essential not only for understanding the life at the molecular level but also for advancing applications in drug discovery and disease treatment. Current methods use CNNs and GNNs to extract information from protein sequences and structures to predict protein function, but we find that these methods often suffer from the inconsistency issue. We will develop a novel reinforcement learning framework to address this problem and achieve accurate predictions.

## 1 Background

Proteins are fundamental biomolecules that perform a wide range of essential tasks in living organisms, from catalyzing biochemical reactions to maintaining cell structure. The unique amino acid sequence of protein determines its three-dimensional structure, which in turn determines its biological function. Protein functions are incredibly diverse and can range from enzymatic activity, structural support, and molecular transport to cell signaling and immune response. Accurately predicting protein function is essential not only for understanding the life at the molecular level but also for advancing applications in drug discovery and disease treatment[1].

Recent advances in high-throughput sequencing have dramatically increased the number of protein sequences in databases, yet the majority remain unannotated due to the high cost of experimental methods. This has increased the demand for more efficient and accurate methods to predict protein functions. Traditional computational methods, such as sequence alignment, transfer functional information from known proteins to unknown proteins. However, more recently, deep learning has achieved great success in this area. We will briefly introduce these methods in the following section.

## 2 Definition

**Protein.** Proteins are composed of sequences of amino acid residues and fold into unique three-dimensional structures via non-covalent side-chain interactions. The protein sequence can be represented as $S = [s_1, s_2, \ldots, s_n]$, where $s_i \in \{0, 1, \ldots, 19\}$ indicates one of the 20 types of natural amino acids. The corresponding structure is denoted as $\mathcal{X} = [x_1, x_2, \ldots, x_m]$, where $x_i \in \mathbb{R}^{L \times 3}$ represents the coordinates of the atoms in the residue.

**Protein Function.** Protein functions are described using the Gene Ontology (GO)[2], which is one of the most successful ontology in biology. GO annotations classify protein functions into three main categories: Molecular Function (MF), Biological Process (BP), and Cellular Component (CC). In each category, GO utilize a Directed Acyclic Graph (DAG) structure to represent the relationships between various protein functions, where the function of each node is encompassed within its parent node. A protein may have multiple GO annotations, reflecting its diverse roles and functions.

38th Conference on Neural Information Processing Systems (NeurIPS 2024).

**Protein Function Prediction.** Protein function prediction can be formulated as a classification problem, which aims to predict the protein function (annotated in GO) given the protein sequence $S$ and its three-dimensional structure $\mathcal{X}$.

## 3 Related Work

With the promising progress in deep learning, numerous methods were developed to predict protein function. These methods can be roughly divided into two categories: sequence-based and structure-based. Sequence-based methods utilized 1D convolutional neural networks (CNNs) or Transformer models to generate specific representations for protein sequences[3, 4]. Later, approaches that combined both protein sequences and homology information showed significant improvements[5, 6]. Recent developments in protein structure prediction allowed researchers to obtain the possible three-dimensional structure of a given protein sequence[7, 8, 9]. Therefore, many structure-based methods utilized graph neural networks (GNNs) and the message-passing paradigm to extract features from protein structural information[10, 11]. Specifically, each residue receives signals from its geometric neighborhood at every layer, and a graph pooling layer then summarizes these representations into a protein-level representation for classification. A newly developed approach, TAWFN, integrated CNNs and GNNs, leveraging both protein sequence and structure information to predict protein function[12].

However, we find that previous methods may suffer from **the inconsistency issue**. Specifically, given that GO annotations use DAG to represent the relationship between different terms, the information contained in a node is strictly included in its parent node. This means that if a protein has the function of a lower-level node, it must also possess the function of its parent node. But existing methods often treat these nodes separately during prediction, leading to situations where a protein is predicted to have the function of a specific node but lacks the function of its parent node.

## 4 Proposed Method

In this study, we adopt the methodology of HiLAP[13], treating the task of protein function prediction as a hierarchical text classification (HTC) problem, and introduce reinforcement learning to address the inconsistency issue.

We initially utilize pre-trained encoders (e.g., ESM-2[14]) to efficiently transform protein sequences and structures into embeddings of fixed feature dimensions, serving as input for subsequent processing steps. We formulate the problem as a Markov decision process (MDP) and utilize reinforcement learning techniques to train a policy network. This network incrementally makes decisions based on the current labeling results, generating new labels or deciding to terminate the labeling process. At the initial state, the labeling node is at the root label. In each training episode, the protein's embedding and the annotated status within the protein functions DAG serve as the state, while all the child nodes of the currently visited nodes in the protein functions DAG constitute the action space. The policy network selects an available action from the dynamically changing action space to continue labeling or to terminate labeling. When the policy network decides to stop labeling, all explored nodes will form the final result of multi-label classification.

We employ robust reinforcement learning methods, such as Proximal Policy Optimization (PPO)[15], to complete the training process. We assess the overall label quality using an example-based F1 (EBF) score and incorporate the changes in this score as part of the reward during each interaction between the agent and the environment. Additionally, we introduce other reward mechanisms to accelerate the training process or enhance training quality. To compensate for potential deficiencies in the embedding space, we also reward the agent's diverse predictions and exploratory behaviors, encouraging it to escape from local optima caused by pre-trained encoders and to regularize these behaviors with appropriate rewards.

We will conduct the experiments using the same dataset as TAWFN[12], which includes 36,639 protein structures from the PDB database and 42,994 protein structures from the AlphaFold protein structure database. These sturctures will be split into training, validation and testing sets in an 8:1:1 ratio. We will compare experimental results with three baseline models including DeepGO-SE[16], HEAL[17] and TAWFN.

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
