# OpenReview forum: "【Proposal】A Reinforcement Learning Framework for Protein Function Prediction"
_tsinghua.edu.cn/THU/2024/Fall/AML — THU 2024 Fall AML Submission_

### Official Review · ~Huajun_Bai1 · 2024-11-07
**Reinforcement Learning for Protein Function Prediction: A Promising Interdisciplinary Approach**

**Rating:** 7
**Confidence:** 3

**Review:**

Strengths

1. Interdisciplinary Approach: The proposal to integrate reinforcement learning into protein function prediction is an innovative step towards leveraging advanced machine learning techniques in bioinformatics. This interdisciplinary approach has the potential to significantly enhance the accuracy and efficiency of predicting protein functions.

2. Addressing Inconsistency in Predictions: The proposal acknowledges the critical issue of inconsistency in protein function predictions and aims to address it through a novel framework. This focus on improving the reliability of predictions is a strength, as it directly tackles a known challenge in the field.

3. Hierarchical Classification Perspective: By framing protein function prediction as a hierarchical text classification problem, the proposal opens up new avenues for leveraging the structured information in Gene Ontology (GO) annotations, which is a sophisticated way to handle the complexity of protein functions.

Weaknesses

1. Limited Discussion on Embedding Space Deficiencies: The proposal mentions the use of pre-trained encoders but does not elaborate on how potential deficiencies in the embedding space might affect the prediction accuracy. Further discussion on how these deficiencies will be identified and mitigated would strengthen the proposal.

2. Scalability and Generalizability: While the proposal outlines a promising method, it does not provide insights into how the approach will scale to the vast and diverse dataset of protein sequences and structures. Addressing scalability and generalizability would be important for assessing the practical application of the proposed framework.

3. Reward Mechanism Design: The proposal introduces various reward mechanisms to enhance training, but it could benefit from a more detailed explanation of how these rewards will be designed to effectively guide the learning process and ensure that the model captures the nuances of protein functions.

---

### Official Review · ~Juncheng_Yu1 · 2024-11-07
**Reinforcement Learning Framework for Protein Function Prediction: A Clear and Promising Approach**

**Rating:** 9
**Confidence:** 3

**Review:**

## Summary

This paper proposes a reinforcement learning framework for protein function prediction, addressing a critical challenge in bioinformatics. By formulating the task as a hierarchical text classification problem and incorporating reinforcement learning, the authors aim to improve prediction consistency for complex protein functions. The research holds great potential to advance protein function prediction methods by integrating hierarchical labeling and RL techniques, which could significantly benefit drug discovery and molecular biology.

## Strengths

- **Clear and Well-Written Presentation**: The paper is well-written and effectively conveys the motivation, background, and methodology. The clarity in the task formulation and method description makes it accessible and engaging for readers.

- **Importance of the Research Problem**: Protein function prediction is a vital area of study with applications in understanding life processes and advancing medical research. This paper addresses a critical and relevant issue, especially considering the rapid growth of unannotated protein data and the high costs associated with experimental validation.

- **Strong Methodological Approach**: The methodology, leveraging hierarchical classification and reinforcement learning, is sound and well-reasoned. The authors' use of hierarchical labeling reflects the inherent structure of protein function relationships and shows promise for enhancing prediction accuracy.

- **Potential for Impactful Results**: The proposed framework is highly relevant and, if successful, could have significant applications in the field. The approach demonstrates a strong potential to overcome limitations seen in existing methods, particularly the issue of prediction inconsistency with hierarchical labels.

## Weakness

- **Clarification on Relation to AlphaFold**: Given the significance of AlphaFold in protein function prediction, it is challenging to discuss advancements in this field without addressing its impact. This proposal does not mention AlphaFold, and providing a clear distinction between this work and AlphaFold’s approach would strengthen the contribution. Highlighting the unique aspects and innovations relative to AlphaFold could clarify how this framework advances protein function prediction in novel ways.

## Score

- **Soundness**: 8/10

- **Contribution**: 8/10

- **Presentation**: 9/10

---

### Official Review · ~Peidong_Zhang1 · 2024-11-08
**Strengths and limitations of proposal**

**Rating:** 8
**Confidence:** 4

**Review:**

This proposal introduces a novel reinforcement learning (RL) framework to address the inconsistency issue in protein function prediction. By treating protein function prediction as a hierarchical classification problem, the approach leverages RL to improve the accuracy of label predictions in Gene Ontology (GO) annotations. The use of pre-trained encoders and reinforcement learning algorithms is a key strength, offering potential improvements in predictive consistency. However, the proposal lacks specific details on how the reinforcement learning approach compares to existing methods in terms of computational efficiency and scalability, especially when dealing with large datasets. Additionally, more clarity on how the system handles potential issues related to data imbalance and missing annotations would be beneficial.

---

### Official Review · ~Lu_Fan_DB1 · 2024-11-09
**Review of the Proposal: "A Reinforcement Learning Framework for Protein Function Prediction"**

**Rating:** 9
**Confidence:** 4

**Review:**

This proposal presents a reinforcement learning framework for protein function prediction, addressing the inconsistency in existing methods that treat Gene Ontology (GO) annotations as separate nodes. By framing protein function prediction as a hierarchical classification task, the authors aim to leverage reinforcement learning to improve prediction accuracy and ensure label consistency.

---

### Official Review · ~Xiaoqian_Liu7 · 2024-11-10
**Clear Problem and Method Statement**

**Rating:** 9
**Confidence:** 3

**Review:**

The proposal "A Reinforcement Learning Framework for Protein Function Prediction" presents a novel method to predict protein functions using a reinforcement learning (RL) framework. The authors aim to address the inconsistency issue in current deep learning approaches for protein function prediction by formulating the task as a hierarchical text classification problem and applying RL techniques.

The proposal is innovative, leveraging pre-trained encoders and reinforcement learning to incrementally make decisions based on the current labeling results, which is a promising approach to handle the hierarchical nature of protein function annotations. The use of robust RL methods like PPO and the incorporation of diverse prediction rewards are well-considered to enhance the training process.

---

### Official Review · ~Nan_Sun10 · 2024-11-11
**A Solid Reinforcement Learning Approach to Enhance Consistency in Protein Function Predictio**

**Rating:** 8
**Confidence:** 4

**Review:**

This paper introduces a novel reinforcement learning  framework for protein function prediction, addressing the inconsistency challenges that traditional deep learning methods often face.

The work stands out by treating the task as a hierarchical text classification problem, where RL is employed to improve the consistency of predictions with the Gene Ontology hierarchy.

This approach is innovative and highly relevant, given the critical importance of protein function prediction in fields like drug discovery and disease treatment. By applying techniques like Proximal Policy Optimization, the proposed framework dynamically adjusts predictions to maintain hierarchical relationships within the GO structure, an aspect overlooked by many current methods.

Evaluation using the TAWFN dataset and comparisons with state-of-the-art methods are anticipated, promising insights into the model's performance against established benchmarks.

However, the RL-based framework may be resource-intensive, posing scalability challenges. The paper could benefit from more clarity on the metrics and benchmarks used for comparison.

---

### Official Review · ~Ruilin_Hu2 · 2024-11-12
**Review of "A Reinforcement Learning Framework for Protein Function Prediction"**

**Rating:** 9
**Confidence:** 4

**Review:**

This proposal introduces an innovative reinforcement learning framework for protein function prediction, addressing inconsistencies in current CNN- and GNN-based methods. By treating protein function prediction as a hierarchical text classification task, the authors utilize reinforcement learning to dynamically assign labels within the Gene Ontology structure. Their approach enhances prediction accuracy by leveraging reward mechanisms and promoting diverse exploratory behavior in the policy network. With rigorous testing against competitive baselines, this method shows strong potential in improving functional annotations, substantiating its high ranking and strong acceptance for contribution to the field.

---

### Official Review · ~Kaiwei_Zhang3 · 2024-11-12
**Proposes a novel and innovative method**

**Rating:** 9
**Confidence:** 3

**Review:**

**1. Summary:**

The proposal outlines a novel approach for protein function prediction using reinforcement learning (RL). It aims to overcome inconsistencies in traditional methods. The authors intend to introduce a reinforcement learning framework that will leverage the hierarchical structure of protein function annotations in the Gene Ontology (GO) to improve prediction consistency.



**2. Clarity:**

The proposal is mostly clear, though some technical aspects of the methodology could benefit from additional detail. There is also room for clearer explanation of terms like "inconsistency issue". The proposal is quite unfriendly to those who are not familiar with Bioinformatics.



**3. Originality:**

The approach is innovative, as it attempts to adapt reinforcement learning for hierarchical text classification within protein function prediction. The use of RL is also quite unique in this specific field.



**4. Significance:**

Protein function prediction is of great importance and has vast implications in molecular biology and bioinformatics. Improved accuracy in this field can benefit areas like drug discovery, disease treatment, and our overall understanding of protein biology.



**5. Pros:**

* **Novel approach.** Introducing reinforcement learning in protein function prediction to address the hierarchical structure of GO annotations is a creative and potentially impactful idea.
* **Solid Foundation in Related Work.** The authors demonstrate a strong understanding of current methods and how they fall short, justifying the need for a new approach.



**6. Cons:**

- **Lack of Clarity.** It would be better to illustrate more explicitly on terms such as "inconsistency issues".
- **Ambiguity in Reward Mechanism.** More details are needed on the reward mechanisms to understand how they will ensure the policy network learns meaningful label assignments.

---

### Official Review · ~Kittaphot_Saengprachathanarak1 · 2024-11-12
**Review of "A Reinforcement Learning Framework for Protein Function Prediction"**

**Rating:** 8
**Confidence:** 4

**Review:**

The proposed paper presents an innovative reinforcement learning (RL) framework for protein function prediction, addressing the inconsistency issue seen in existing methods. The authors effectively combine hierarchical text classification and reinforcement learning techniques, such as Proximal Policy Optimization (PPO), to enhance predictions by modeling protein function within the Gene Ontology (GO) structure. Their approach utilizes pre-trained encoders and introduces reward mechanisms to encourage diverse predictions and avoid local optima. While the methodology is novel, further empirical validation through experiments and comparisons with baseline models like TAWFN is needed to assess its practical performance. The clarity and organization of the paper are strong, but the absence of empirical results slightly limits its current impact. Overall, this work promises a significant contribution to protein function prediction using reinforcement learning.